# Interface Behavior of Asphalt Pavements Constructed by Conventional and Double-Decked Paving Methods

**DOI:** 10.3390/ma13061351

**Published:** 2020-03-17

**Authors:** Ke Mu, Zhiwei Gao, Xin Shi, Yanwei Li

**Affiliations:** 1State Key Laboratory of Road Engineering Safety and Health in Cold and High-Altitude Regions, CCCC First Highway Consultants Co., LTD, Xi’an 710065, China; muk@ccroad.com.cn; 2XiZang Key Laboratory of Optical Information Processing and Visualization Technology, XiZang Minzu University, Xianyang 712082, China; gzw@xzmu.edu.cn; 3Department of Civil Engineering, Hebei Jiaotong Vocational and Technical College, Shijiazhuang 050035, China; 4Hebei Provincial Communications Planning and Design Institute, Shijiazhuang 050011, China; yanweili2020@yeah.net

**Keywords:** Double-decked paving, conventional paving, interface strength, direct shear test, uniaxial tensile test, trial road

## Abstract

Asphalt pavement consists of multiple layers of asphalt with a progressive decrease of nominal maximum aggregate size from the bottom to the top, which can be constructed by the double-decked or the conventional paving method (i.e., layer by layer). Reliable interface strength between the fine- and the coarse-grained layer of asphalt mixture is prerequisite to ensure the serviceability of the asphalt pavement. This study aims to compare the interface behavior of the asphalt pavement constructed by the conventional and the double-decked paving methods through laboratory and trial pavement tests. Laboratory test results show that the interface strength of the specimen prepared by the double-decked paving method is mainly contributed by the interlocking of the coarse- and the fine-grained asphalt mixture, fundamentally different from the conventional paving method, in which the interface strength is mainly provided by the tack coat oil. More importantly, the interface shear strength and the uniaxial tensile strength of specimens prepared by the double-decked paving method are about 1.5–1.8 times larger than that of specimens prepared by the conventional paving method. To verify the applicability of laboratory experimental findings to the practical engineering, a trial road was paved in situ using both double-decked and conventional paving methods. Cored specimens were collected from the trial road and their interface strengths are tested. Comparisons of the interface strength obtained from cored specimens further prove that the asphalt pavement constructed by the double-decked paving method has larger interface strength than that of the asphalt pavement constructed by the conventional paving method.

## 1. Introduction

The double-decked paving method can pave and compact two layers of asphalt mixture simultaneously during the construction of an asphalt pavement [1,2]. In this method, coarse- and fine-grained asphalt mixtures are added to the special paving equipment, which has two hoppers and two independent paving systems. Thereafter the equipment paves two different asphalt mixtures simultaneously through the following three steps: First, during the paving process, the lower layer of asphalt mixture is compacted by the tapping and shaking devices, which were installed in paving equipment. Second, the upper layer of asphalt mixture is paved on the lower pavement layer. Third, both the upper and lower layer of asphalt mixture are compacted simultaneously with the paving equipment. As shown in Figure 1a, the double-decked paving method enables the upper and lower layers of asphalt mixture to be interlocked with each other after the heavy compaction, thus greatly improving the interface strength of the asphalt pavement. In contrast to the double-decked paving method, shown in Figure 1b, the conventional method paves the asphalt mixture layer by layer and the interface strength is mainly provided by the tack coat oil. According to the literature [3,4], asphalt mixture, pavement characteristics, temperature and compaction level would also affect the interlayer strength. In practical engineering, researchers proposed alternative tack coat oils with advanced bonding properties to improve interlayer strength [5]. However, it is still difficult to have reliable interface strength for the asphalt pavement using tack coats with excellent bonding properties. For conventional paving method, the tack coat and upper layer of asphalt mixture can be paved when the temperature of lower layer of asphalt mixture falls to ambient temperature, which may need 1–3 h based on the atmospheric temperature [6,7]. It should be noted that mixing and transporting equipment need to be readjusted because of different gradations of asphalt mixture for the upper and lower pavement. In practical engineering, the upper layer of asphalt mixture is usually applied 24 h later after the construction of the lower layer of asphalt mixture. During the suspension, the lower layer of the asphalt mixture acts as the workbench for the construction of the upper layer. An uneven distribution of the tack oil and pollutions (e.g., soil and fuel leaked from vehicle) may lead to a weakness interface layer, which has a low bonding strength. On the other hand, as two layers of asphalt mixtures are paved simultaneously at relatively high temperatures, the construction efficiency for the double-decked paving method improves significantly compared with the conventional paving method. 

The interface strength is important for the serviceability of the asphalt pavement [8,9,10]. At the slope, junction, and small curvature of the pavement, a large horizontal force would be produced within the pavement due to vehicles climbing, frequent starts and stops, and sharp turns [11,12,13]. Thus, road damage, e.g., crack and rut, can be easily caused in the case of a pavement without reliable interface strength [14,15,16]. In addition, the elastic layering theory has been widely adopted to design asphalt pavement based on the calculated stress distribution and deformation behavior under traffic loads [17]. The presuppositions of the elastic layering theory include continuity and small deformation of material [18,19]. An uneven distribution of tack oil and pollution (e.g., soil and fuel leaked from vehicles) may lead to the weakness interface layer, which has low bonding strength. If a weakness layer or separation between the upper and layer of asphalt mixture occurs in the asphalt pavement, the pavement design based on the elastic layering theory would induce significant errors [20,21,22]. Due to above reasons, a reliable interface strength is significant for ensuring the serviceability of asphalt pavement and assuring that the actual state of asphalt pavement is consistent with the design state using the elastic layering theory.

This study aims to investigate the interface behavior of asphalt pavement constructed by the double-decked and the conventional paving methods. The processes of double-decked and conventional paving were simulated through Marshall compaction experiment to determine the interlocking of the upper and lower layers of the asphalt mixture. Through direct shear and uniaxial tensile tests, the interface strength of specimens prepared by the double-decked and the conventional paving methods are compared and analyzed. Finally, a trial road was paved with the double-decked and the conventional paving method, and the cored specimens were collected from the trial road for further testing of the interface strength by direct shear and uniaxial tensile tests.

## 2. Materials and Test Methods

### 2.1. Specimens Preparation

The asphalt pavement constructed by the conventional and the double-decked paving method is simulated through Marshall compaction experiment. For the upper layer of asphalt mixture, the gradation type is AC-13, and the aggregate material is basalt. The mass ratio of asphalt to aggregate is 4.5%, while the mass ratio of filler to aggregate is 5.3%. For the lower layer of asphalt mixture, the gradation type is AC-20, and the aggregate material is basalt, similar to the upper layer of asphalt mixture. The ratio of asphalt to aggregate mass is 4.7%, and the ratio of mineral powder to aggregate mass is 5.5%. The particle size distribution of aggregate in the AC-13 and AC-20 are shown in Figure 2. In addition, 70# asphalt (Shell (China) Limited, Beijing, China) was used in the test. The property of asphalt and aggregate are shown in Table 1 and Table 2.

Iron hoops (Yaxing Equipment of Civil Engineering Limited, Xi’an, China) with an internal diameter of 101.6 mm, a height of 65.5 mm, and a thickness of 4 mm were adopted for the compaction of specimens. The two gradated asphalt mixtures were compacted into the iron hoop, and the Marshall compactor (Yaxing Equipment of Civil Engineering Limited, Xi’an, China) was used for 75 blows of Marshall compactor to obtain the tested specimens [23,24]. Since the density of the asphalt mixture ranges from 2.35 to 2.45 g/cm^3^ because of different gradations, 5% volume of the container was reserved for the mould release and specimen compaction. The total weight of each specimen is 1200 g, with each of the mass of gradation mixture equaling to 600 g. The specimen thickness is usually 2.5–4.0 times larger than the nominal maximum size of asphalt mixture, in order to prevent interaction between aggregates during compaction. Due to the limited size of the Marshall test specimen mold, this criteria may not be satisfied for the preparation of coarser asphalt mixture. However, the advantage of using Marshall test specimen mold is that the compaction criteria are widely recognized. In this study, coarse aggregates are deliberately distributed in the mold to ensure an even distribution, and hence reduce their interactions during compaction.

For the conventional paving method, the upper and lower layers of asphalt mixture were paved layer by layer, with the tack coat oil sandwiched between the two layers. First, 600 g of AC-20 asphalt mixture was added into the iron hoop. After 75 compactions, the asphalt mixture was cooled to the room temperature of 20 ℃, Styrene-Butadiene-Styrene (SBS) modified asphalt was spread on the surface of the first layer as the bonding material. Finally, 600 g of AC-13 asphalt mixture was added into the iron hoop, and the mixture was applied with another 75 compactions. Finally, the prepared specimen was cooled to the room temperature of 20 ℃. It should be noted that the temperature of the lower layer should fall to 20 °C before applying the tack coat during specimen preparation with the conventional paving method. Thereafter, the upper layer was compacted at 160 °C.

For the double-decked paving method, two graded asphalt mixtures were paved and compacted simultaneously. After adding 600 g of AC-20 asphalt mixture into the iron hoop, the first 30 blow compactions and smoothing were performed. Subsequently, the 600 g of AC-13 asphalt mixture was added into the iron hoop, and the specimen was obtained by applying 75 blows of Marshall compactor. For Marshall compaction, the first 30 blows and smoothing are performed to simulate the compaction of lower layer of asphalt mixture by tapping and shaking device in the double-decked paving method. The compaction of both upper and lower layer of asphalt mixture by the compaction equipment in the double-decked paving method is simulated by the 75 blows of Marshall compactor. For double-decked paving method, lower and upper layer of asphalt mixture were paved under the same temperature. It should be noted that the temperature of the asphalt mixture was controlled at 160, 140, and 120 °C during the specimen compaction, aiming to make three groups of specimens for studying the effects of compaction temperature on interface behavior of asphalt pavement.

Six groups of samples were prepared and tested under conventional method, with each of 3 groups was prepared in laboratory and on-site coring. On the other hand, twelve groups of samples were prepared by the double-deck method, with nine groups made in laboratory and three groups obtained through on-site coring. Quantification of interlocking depth, direct shear tests and tensile tests were carried out through laboratory tests. Each test was carried out three times, using three samples to obtain the average values. Summary of test program was shown in Table 3.

### 2.2. Test Methods

#### 2.2.1. Quantifying the Interlocking of Upper and Lower Layers of Asphalt Mixture

To give a quantitative analysis the interlocking, aluminum slices were placed between the upper and lower layers of asphalt mixture (i.e., AC-20 and AC-13). The aluminum slice is very soft and can be deformed during compaction once the two layers are compressed against each other. After compaction, the interlocking depth between upper and lower layers can be quantified by measuring the deformation of the slice. The following detailed procedures were observed to quantify the interlocking depth in this study:(a)AC-20 was put into the iron hoop and 75 blows of Marshall compactor were applied to simulate the conventional paving method in practical engineering. For the double-decked paving method, 30 light compactions were applied to the specimen to obtain a smooth surface.(b)An aluminum slice with an internal diameter of 101.6 mm and a thickness of 0.1 mm was placed on the top of the lower layer of asphalt mixture.(c)The AC-13 mixture was added into the iron hoop and compacted 75 times from top and bottom sides. The conventional and double-decked paving method share a similar procedure for this step.(d)The specimens were separated along the middle plane, and the aluminum slice was removed, as shown in Figure 3. The middle plane was selected as the reference plane for the quantification of interlocking depth. Ten points were selected along the radial direction of the aluminum slice. The average height of the convex and concave was used to quantify the interlocking depth. The measurements were abandoned when their values are larger than 20% of the average value, and then the average value was determined again with left measurements as the value of interlocking depth.

#### 2.2.2. Direct Shear Apparatus

Direct shear tests were carried out to measure the interface shear strength between upper and lower layers of asphalt mixture under the conventional and the double-decked paving method [25,26]. As shown in Figure 4, the test apparatus (Yaxing Equipment of Civil Engineering Limited, Xi’an, China) consists of a shear box, a loading rod, a load cell and a motor. During the test, the motor pushed the loading rod to move the lower part of the shear box while the position of the upper part of the shear box was fixed. Therefore, a shear force was applied to the interface between the upper and lower parts of the shear box which contained the tested specimen. In this study, the shearing was applied with a speed of 10 mm/min until the specimen was destroyed. During the shearing, the load cell continuously measured the shear force. The measured shear force was divided by the cross-sectional area of the specimen to obtain the interface shear stress. Each test was conducted with three independent specimens, and the average value of the maximum shear stress was taken as the value of interface shear strength. During shearing, the horizontal deformation of specimen was recorded through the strain gauge. It should be noted that the dilation of specimen was allowed during the test. Unfortunately, the vertical deformation of specimen was not measured, and hence the dilatancy cannot be quantified in this study. In addition, the nominal area of sample was used to calculate the shear stress.

#### 2.2.3. Uniaxial Tensile Apparatus

To compare the tensile strength of the asphalt pavement constructed by the conventional and double-decked paving methods, the prepared specimens were tested by uniaxial tensile tests (Yaxing Equipment of Civil Engineering Limited, Xi’an, China) [27,28]. The epoxy resin and curing agent used in the test are bisphenol oil-based epoxy resin, and low molecular polyamide, respectively. The epoxy resin and the curing agent were cured for 72 h. As shown in Figure 5, the top and bottom surface of the specimen were bonded to top cap and concrete slab, respectively. After ensuring a firm bond, the tensile force was applied through the loading rod. The loading rod pulled the test specimens upward with a speed of 10 mm/min until a crack was observed in the tested specimen. The tensile stress of the specimen was calculated by dividing the tensile force by the cross-sectional area of the specimen.

## 3. Experimental Results

### 3.1. Interlocking Depth

Table 4 shows the measured interlocking depth of specimens prepared by the conventional and the double-decked paving method. It is clear that the interlocking depth of the specimens prepared by the conventional paving method is very small (i.e., 0.6 mm). The interface strength of the specimen prepared by the conventional paving method is mainly provided by the tack coat oil. More importantly, the interlocking depth of the specimens prepared by the double-decked paving method is 7–10 times larger than that of the specimen prepared by the conventional paving method. On the other hand, the interlocking depth between the two layers of asphalt mixture is highly dependent on the compaction temperature under the double-decked paving technology. The higher the compaction temperature is, the higher the interlocking depth is. With compaction temperatures increasing from 120 ℃ to 160 ℃, the interlocking depth increased by 46.3%. There is no tack coat oil between the two asphalt mixture layers for specimens prepared by the double-decked paving method, and the interface connection is completely contributed by the interlocking of the upper and lower layers of asphalt mixture.

### 3.2. Interface Shear Strength

Figure 6 and Figure 7 show the results of direct shear test for specimens prepared by the conventional and double-decked paving technology, respectively. As shown in the Figure 6, the shear stress first increases non-linearly with increasing horizontal displacement and reaches a peak value. With further increasing in horizontal displacement, the shear stress decreases abruptly. Under double-decked paving, aggregates interlock with each other for the specimens prepared with double-decked paving method. After reaching the peak value, the shear stress does not decrease immediately with an increase in horizontal deformation, mainly because that some stones stuck between the upper and lower layers. It should be noted that all tested specimens show similar patterns of stress-strain relationship. The interface shear strength of the specimen prepared by the conventional paving method is 0.49 MPa, and the maximum shear strength appears at the horizontal displacement ranging from 4.9 to 5.7 cm, with the average value equaling to 5.3 cm. More importantly, the interface shear strength of specimens prepared by the double-decked paving method is 1.5–1.8 times larger than that of specimens prepared by the conventional paving method. On the other hand, the maximum shear strength occurs at the displacement ranging from 5.7 to 6.6 cm, with the average value equaling to 6.2 cm. It is clear that the ductility of the double-decked paving method is 17.0% higher than that of the conventional paving method. In addition, under the double-decked paving method, the interface shear strength and ductility are highly dependent on the compaction temperature of asphalt mixture. Increasing the compaction temperature from 120 °C to 160 °C, the interface shear strength and the ductility increased by 17.6% and 12.2%, respectively. The improvement of strength and deformation behavior of specimens compacted at different temperatures coincides with the conclusion that the higher the compaction temperature of asphalt mixture, the greater the interlayer interlocking depth. It should be noted that high compaction temperatures also have negative effects on the bonding quality of asphalt mixture. In practical engineering, the mixing temperature of asphalt mixture is generally lower than 180 °C.

### 3.3. Interface Tensile Strength

Table 5 shows comparisons of the tensile strength of the specimens prepared by the conventional and double-layer paving method. For the conventional paving method, the interface tensile strength is only 0.53 MPa which is mainly attributed to the tack coat oil. For the double-decked paving method, the interface tensile strength is 1.6–1.8 times larger than that of the specimens prepared by the conventional paving method, because the tensile strength is contributed by the interlocking of the upper and lower layers of asphalt mixture. In addition, for the double-decked paving method, the interface tensile strength is highly dependent on the compaction temperature of the asphalt mixture. When the compaction temperature is elevated from 120 °C to 160 °C, the interface tensile strength was increased by 12.6%. This is consistent with the conclusion that the higher the compaction temperature, the greater the interlocking of the upper and lower layers of asphalt mixture.

## 4. Field Test from the Trial Road

To verify the applicability of laboratory test to practical engineering, the trial road for Shangzhou to Luonan Highway in Shaanxi Province (K10 + 975-K12 + 795), China was constructed using both conventional and double-decked paving method. For the trial pavement, the gradation of the asphalt mixture was consistent with the specimens tested in the laboratory, with AC-13 and AC-20 for upper and lower layers of the asphalt pavement respectively. For both conventional and double-decked paving methods, the mixing and compaction temperatures of asphalt mixture were 173 °C and 163 °C, respectively. The differences in these two temperatures are attributed to the temperature loss during transportation of asphalt mixture from mixing station to construction site. The dosage of the tack coat used in this study is 0.5 L/m^2^. Layer thicknesses, compaction details were given in Table 6. Following the compaction method in Table 6, the compaction degree of asphalt under conventional and double-deck paving method can reach to 99.2%–99.5%. The effect of air voids on interface strength can be ignored. The test specimens were obtained through on-site coring, as shown in Figure 8. A micrometer was used to measure interlocking depth from the outer edge of the specimen following the procedure proposed in Section 2 and direct shear and uniaxial tensile tests were carried out in a way consistent with those of laboratory tests. Figure 8b shows that a few of air voids can be found in the specimens which were obtained through on-site coring, proving good compaction conditions in this study. This is consistent with the above experimental results that the compaction degree of asphalt can reach to 99.2%–99.5%.

Figure 8b shows that, under conventional paving, the failure only occurs in the location of tack coat, with a flat and smooth mode. Under double-deck paving, the crack first appears in the interface and then extends to the lower layer. This is mainly because of the higher bonding strength provide by interlocking of upper and lower layer of asphalt mixture.

Table 7 shows the comparisons of interlocking depth, interface shear strength, and tensile strength of the tested specimens prepared in the laboratory and collected from the trial road. The results obtained from the cored specimens show that the interface shear strength and tensile strength of the pavement are twice larger than that of the conventional pavement under the double-decked paving method. On the other hand, comparing with the tested specimens prepared in the laboratory, the interlocking depth, shear strength and tensile strength of the sample collected from the trial pavement are 26.7%, 32.9% and 16.3% larger, respectively. This is due to the fact that, on the one hand, the compaction temperature of the asphalt mixture reached 163 °C in the trial road which is larger than the compaction temperatures in the laboratory (i.e., 160 °C). As mentioned before, the high temperature is beneficial to enhance the interface strength. On the other hand, the high frequency vibratory roller was used in the construction of trial pavement and the compaction energy is much higher than that of laboratory Marshall compactor. Thus, the interface strength of the specimens collected from the trial road is higher than that of specimens prepared in the laboratory.

## 5. Conclusions

By means of laboratory tests and the construction of a trial road in situ, this paper compares and studies the interface behavior of asphalt pavement constructed by the double-decked and the conventional paving method. The underlying mechanisms of the interface strength for the double-decked paving method are revealed. Further, the influence of the compaction temperature of asphalt mixture on the interlocking depth, shear strength, and tensile strength of the asphalt pavement constructed by the double-decked paving method is analysed. The main conclusions of the study are as follows:(1)For the conventional paving method, the interface strength is mainly provided by the tack coat. When failure occurs, the failure section is located in the tack oil layer. For the double-decked paving method, the interface strength is mainly provided by the interlocking of the upper and lower layers of asphalt mixture. When failure occurs, the failure section extends to upper or lower layer of asphalt mixture.(2)For the double-decked paving method, the interface shear strength and tensile strength are 1.5–1.8 and 1.6–1.8 times larger than that of the conventional paving method, respectively. The average horizontal deformation at failure is 5.3 cm and 6.2 cm under conventional and double-deck paving method, respectively. Compared with the conventional paving method, the ductility (the horizontal deformation at failure) is improved by 17% for specimens prepared by the double-decked paving method.(3)The test results for the trial road are consistent with the laboratory test, which indicates that the double-decked paving method can enhance the interface strength of asphalt pavement not only under the laboratory conditions, but also in terms of practical engineering. Moreover, the interlocking depth, shear strength, and tensile strength measured by cored specimens from the trial road are higher than that in the laboratory test, mainly because of the higher compaction temperature and larger compaction efforts in the field.

## Figures and Tables

**Figure 1 materials-13-01351-f001:**
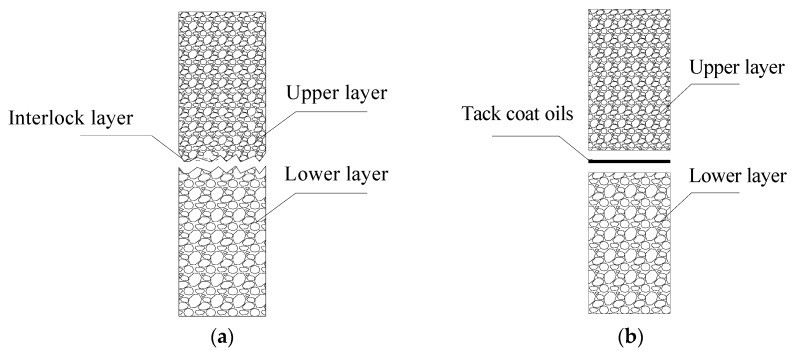
Illustration of the interface in the asphalt pavement constructed by: (**a**) double-decked paving method (**b**) conventional paving method.

**Figure 2 materials-13-01351-f002:**
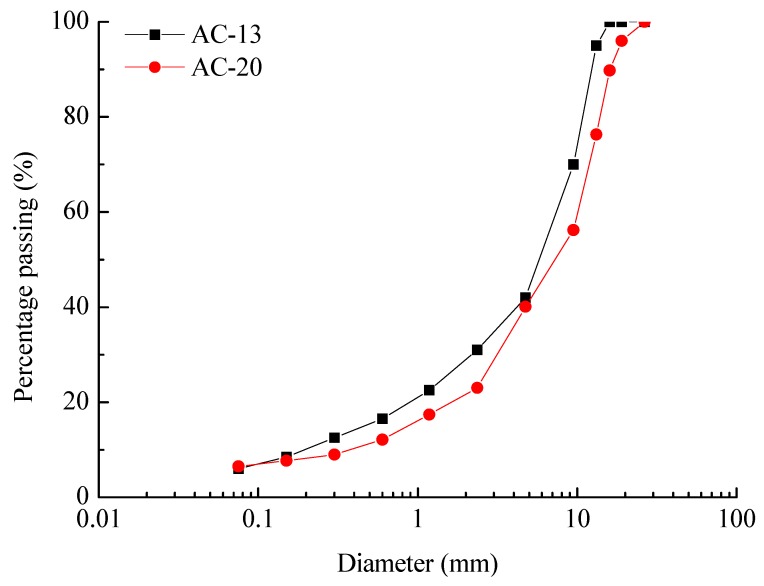
Particle size distribution of the aggregate in the AC-13 and AC-20.

**Figure 3 materials-13-01351-f003:**
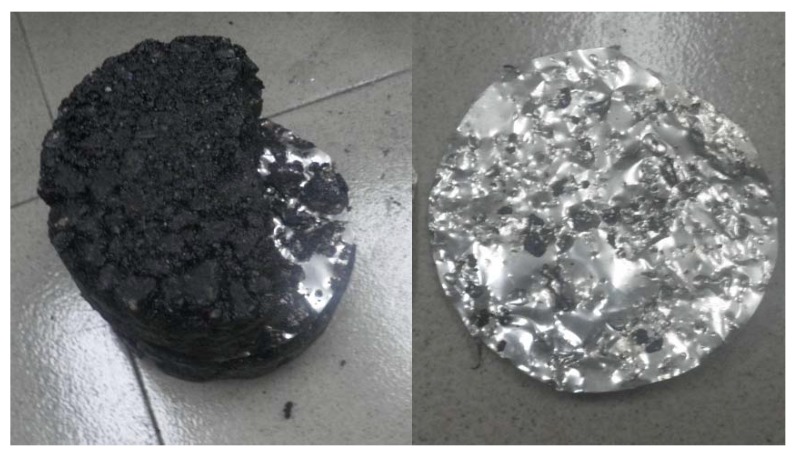
Illustration of the aluminum slice between upper and lower layer of asphalt mixture.

**Figure 4 materials-13-01351-f004:**
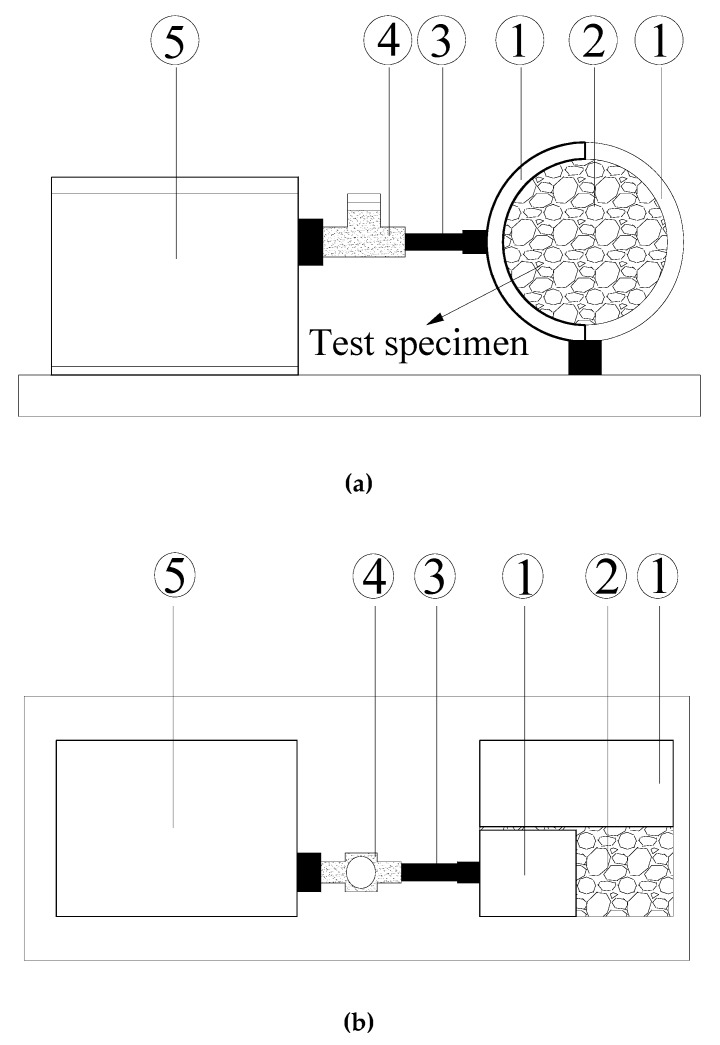
Schematic diagram of the direct shear apparatus: (**a**) Cross-sectional view; (**b**) Plane view. ① shear box ② test specimen ③ loading rod ④ load cell ⑤ motor.

**Figure 5 materials-13-01351-f005:**
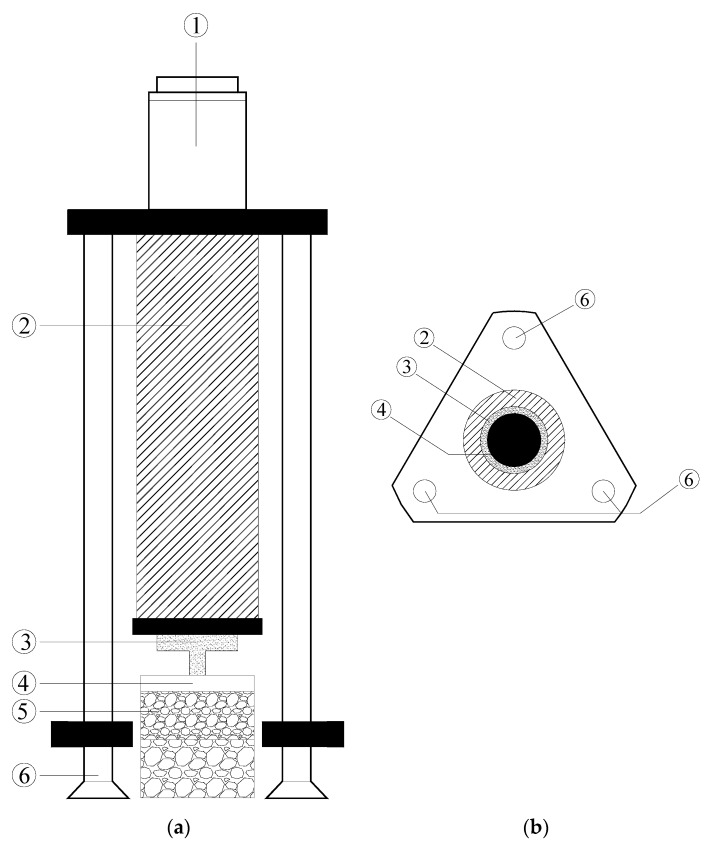
① motor ② loading rod ③ load cell ④ top cap ⑤ test specimen ⑥ loading frame. Schematic diagram of the uniaxial tensile apparatus: (**a**) cross-sectional view; (**b**) plane view.

**Figure 6 materials-13-01351-f006:**
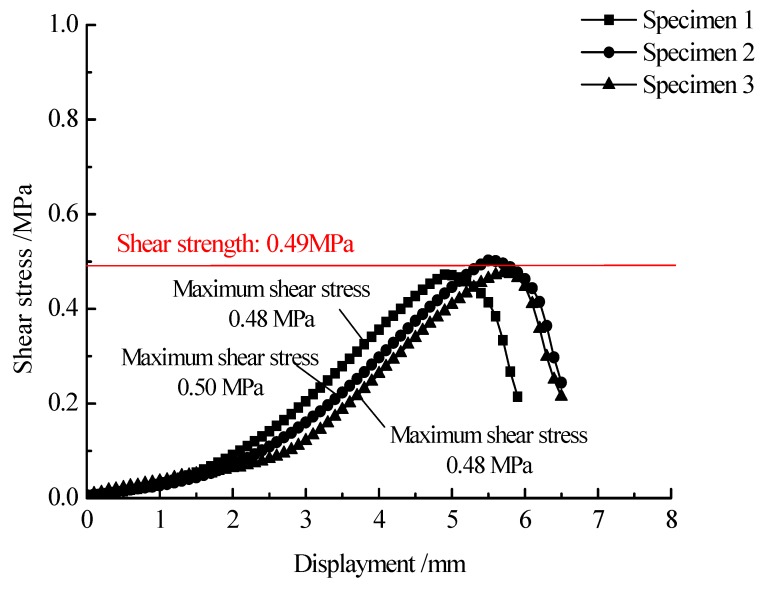
Stress-strain relationship of specimens prepared by the conventional paving method.

**Figure 7 materials-13-01351-f007:**
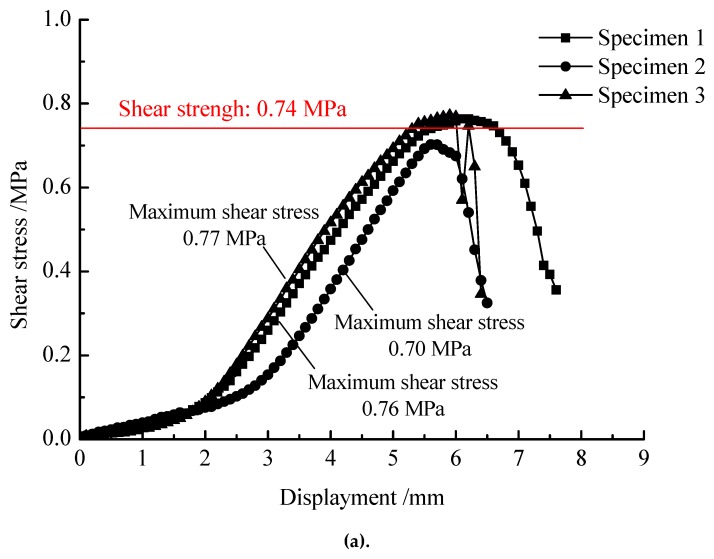
Stress-strain relationship of specimens compacted at different temperatures by the double-decked method: (**a**) 120 °C; (**b**) 140 °C; (**c**) 160 °C.

**Figure 8 materials-13-01351-f008:**
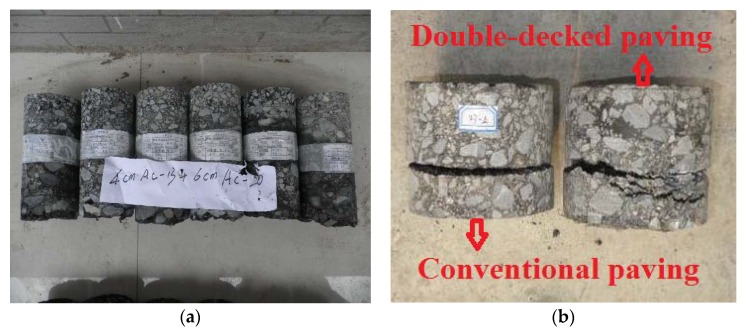
Specimens collected from the trial road constructed by conventional and double-decked paving method: (**a**) before direct shear test; (**b**) after direct shear test.

**Table 1 materials-13-01351-t001:** Properties of 70# asphalt.

Penetration (25 °C)	Ductility (15 °C)	Softening Point	Solubility
7.2 mm	105 cm	50 °C	99.7%

**Table 2 materials-13-01351-t002:** Properties of aggregate.

Crushed Stone Value	Wearing Stone Value	Apparent Specific Gravity
22%	25%	2.7

**Table 3 materials-13-01351-t003:** Summary of test program.

Paving Method	Sample Numbers		Test Item
	Laboratory	Trial Road	
Conventional	3 groups (3 samples for each group)	3 groups (3 samples for each group)	① Interlocking depth ② Direct shear test ③ Tensile test
Double-decked	9 groups (3 samples for each group)	3 groups (3 samples for each group)	

**Table 4 materials-13-01351-t004:** Experimental result for the interlocking depth of the upper and lower layers of asphalt mixture under conventional and double-decked paving method.

Paving Method	Compaction Temperature ℃	Interlock Depth (mm)	Average Value (mm)
Conventional paving method	160	0.3	0.6
		0.4	
		1.1	
Double-decked paving method	120	4.3	4.1
		3.8	
		4.2	
	140	4.4	4.7
		4.6	
		5.1	
	160	6.2	6.0
		5.7	
		6.1	

**Table 5 materials-13-01351-t005:** Tensile strength of specimens prepared by conventional and double-decked paving methods.

Paving Method	Compaction Temperature (°C)	Tensile Strength (MPa)
Conventional paving method	160	0.53
Double-decked paving method	120	0.87
	140	0.92
	160	0.98

**Table 6 materials-13-01351-t006:** Layer thickness and compaction method under conventional and double-decked paving methods.

Thickness	Compaction under Double-Decked Paving	Compaction under Conventional Paving
4 cm AC-13 + 6 cm AC-20	Lower layer was compacted by the tapping and shaking devices. 2 times rolling with 4-ton steel roller. 4 times rolling with 13-ton steel roller. 6 times rolling with 31-ton rubber roller. 2 times rolling with 13-ton steel roller.	4 times rolling with 13-ton steel roller 6 times rolling with 31-ton rubber roller 2 times rolling with 13-ton steel roller

**Table 7 materials-13-01351-t007:** Comparison of interface strength of specimens obtained from the laboratory and the trial road.

Interface Strength	Conventional Paving Method		Double-Decked Paving Method	
	Laboratory	Trial Road	Laboratory	Trial Road
Interlocking depth (mm)	0.6	0.6	6	7.6
Shear strength (MPa)	0.49	0.51	0.87	1.09
Tensile strength (MPa)	0.53	0.54	0.98	1.14

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
