# Peer review of "Interface Behavior of Asphalt Pavements Constructed by Conventional and Double-Decked Paving Methods"

_materials, 2020, doi:10.3390/ma13061351_

Round 1
Reviewer 1 Report
Dear authors,
the paper shall be read by a native speaker or a English professional to avoid some of the errors or unclear formulations. Some of the problems I have identified:
1) pg. 2, line 6: word "behavior" - please, correct
2) pg. 3, first paragraph: check "tack coat". There are minimum two times errors and word "track" is used.
3) provide on page 3 please, explanation what does it mean "advanced tack coat", and why there was 24 hours waiting period after tack coat was applied?
4) pg. 4, line 7: "...if a weakness layer occurs". What does it mean weakness layer?
5) pg. 5 and 6: "75 positive and negative compactions". Usually we use the wording 75 blows of Marshall compactor. Please, check and use correctly in the text.
6) pg. 5: it is written that 600 gram of mixture was used for each layer. Is it not a bit too small amount. I understand the reason (Marshall test specimen molds), but in my opinion it is not correct since you will produce about 3-3.5 cm thick specimen which especially for the coarser asphalt mixture is extremely low.
7) Tab. 1: the compaction temperature for conventional solutions is given with 20°C. How were the specimens compacted at this temperature? Or is it temperature at which the two layers were put together? Even this is not really correct since normally the lower layer shall be compacted, than in lab tack coat applied and finally the upper layer shall be placed on the applied tack coat and compacted. I understand you did it separately and put the two specimens (one for coarse mix and one for fine grained) together. It is questionable how much this correspond to the real application.
8) pg. 11, last sentence: This is true but it has some upper limit and this shall not be ignored. If the temperature is too high the quality of the bitumen can be negatively affected and this will have negative influence on the quality of bond as well.
9) pg. 15: compaction temperature for the field application was 173°C? Is it not too high? Was it the temperature for main compaction phase? Why the temperature was as high and what was then the mixing temperature?
10) conclusions, bullet point (1): "....layers does not occur the interlocking". Please check English in this sentence. The verb "occur" is somehow strange here.
Reviewer 2 Report
GENERAL COMMENTS
The paper presents a laboratory and field study investigating the interface shear properties of double layered asphalt samples prepared with the double-decked method (i.e. two layers paved simultaneously at relatively high temperature).
The topic of the research is of a great interest, but the adopted approach and the related results are affected by serious flaws.
The relevant literature is not taken into account. In particular, the double-decked paving method (core of the research) is not explained in details and the only references cited in the text are in Chinese. If English references do not exist, this paving method should be fully explained providing as much details as possible in order to properly appreciate and interpret the following research approach.
The objective of the paper is clearly stated. However, materials and methodologies are poorly presented since important details are missing. A colleague cannot reproduce the experiments and get the same outcomes based on the available text.
Marshall compaction which provides for the application of a certain number of blows in both upper and lower specimen surfaces does not seem appropriate to simulate compaction of double-layered samples. Moreover, for the conventional paving methods 75 + 75 + 75 + 75 blows are applied to compact the double layered specimens whereas only 75 + 75 blows (plus a not specified light compaction) are applied for the double-decked paving methods for specimens having the same final dimensions (very different compaction energy per unit of volume).
Since the double-decked paving methods could lead to inadequate compaction of the thick double layered surface, specific attention should be devoted to this aspect (air voids of both layers, compaction profile, etc.) while no information is provided in this paper.
Sometimes, results are not properly justified/commented but only presented.
The conclusions are affected by the abovementioned issues.
The use of English should be improved.
SPECIFIC COMMENTS
Please check “tack” instead of “track”
It is not fully correct asserting that an asphalt pavement consists of a fine- and a coarse-grained layer of asphalt mixture. Often, there are three or more asphalt layers with a progressive decrease of nominal maximum aggregate size from the bottom to the top
It is not fully correct asserting that in the conventional paving method the interface strength is mainly provided by the tack coat. Other contributions can also be detected (see, for example, Ferroti et al. ADVANCED TESTING AND CHARACTERIZATION OF INTERLAYER SHEAR RESISTANCE, TRR 2005)
It is not fully correct asserting that it is difficult to have reliable interface strength using conventional paving methods. When proper constructed, adequate (acceptable) interface properties are achieved.
Usually, the tack coat is applied on the cold lower layer and then the upper layer can be paved after only some hours (1-3) depending on the climate conditions
Section 2.1: with “mineral powder” do you mean the filler (particle size lower than 0.063 mm)?
What about the main properties of the aggregates and the bitumen? Are they the same for the upper and the lower layers? Are the mixtures prepared in the lab or taken from the plant? How are the mixtures prepared? What is the dosage of the tack coat? And its properties? Etc.
The Marshall mould should have an internal diameter of 101.6 mm
What does it mean “30 light compactions and smoothing”? Please be more specific using a scientific approach to describe the adopted compaction method. Is it consistent with field compaction?
In case of double-decked paving method, what was the temperature of the lower layer when the compaction of the upper layer started? Is it representative of the compaction in the field?
The height of the convex and concave of the aluminum slice with respect to what? Ten measurements taken where? Please provide more details about the methodology to quantify the interlocking. A colleague should be able to reproduce the method based on the text.
Direct shear apparatus description: since the methodology is not standardized, please provide as much details as possible (is the dilatancy allowed? Are the vertical and horizontal deformations measured? Which is the gap at the interface? For the calculation of the shear strength, do you consider the effective area or the nominal area? Etc.
Page 8: what are the epoxy resin and the curing agent? Could this part be moved further on?
Page 8: how was applied the tensile force? Is it a load controlled or a deformation controlled test?
Figure 5: and the concrete slab?
Table 1: the reported compaction temperatures refer to the lower or the upper layer? And the compaction temperature of the other layer?
Section 3.2: the displacement is in mm, not cm
What do you mean with “anti-deformation properties”? The deformation at failure provides an idea of the ductility of the system
Fig. 6: use the same x and y scales of Fig 7
Fig. 7a: what does it happen to specimen 3?
Table 2: see comment for Table 1.
Section 4: what is the reference plane with respect to interlocking is measured? Be more specific.
Section 4: 173 °C is the laying temperature of the lower or the upper layer? And the temperature of the other layer? Please specify for both paving methods
Section 4: layer thicknesses? Compaction details? Tack coat? Construction phases and times? Etc.
Figure 8b shows that the failure is not at the interface but within the lower layer in the case of the double-decked specimen. Any comment?
Caption of Fig 8: before and after which test?
Table 3: 0.87, not 0.82
Conclusions, point (1): please check
Conclusions, point (2): see comment on the anti-deformation properties
References 10, 12 and 30 are the same
References 21 and 22 are the same
Reviewer 3 Report
This paper aims to investigate the interface behavior of asphalt pavement constructed by the double-decked and the conventional paving method. The work ir very wide and shallow, it presents quite weak data and I believe its scope and scientific contribution are limited.
Some specific comments:
1. The introduction is "studentlike". Citing many sources in one short sentence is not scientifically correct, e.g.: …"The interface strength is important for the serviceability of asphalt pavement [9-12]." In addition, the scientific level of lot of sources and journals is questionable, it is difficult to find when the DOI number is not specified.
2. One general comment is that the author in some places only presenting the test results without rational and strong scientific discussion, the results are not presented systematically. It is unclear how long the experimental investigation lasted and when it began. How many samples were taken from the pavement and were made in the laboratory? A table summarizing the experiment design is required.
3. What do the author refer by saying that “...When the compaction temperature was elevated from 120℃ to 160℃, the interface tensile strength was increased by 12.6%. This is consistent with the conclusion that the higher the compaction temperature, the greater the interlocking of the upper and lower layers of asphalt mixture..." in this work? Is this a reasonable conclusion from several speciments? What about interlocking if the temperature of compaction will be 170, 180, 250℃???
4. The conclusions are too global and should be more specific. It should be clearly highlighted and separated the results of authors experiments. The term “Compared with the conventional paving method, the anti-deformation capacity improves by 1.1-1.2 times for specimens prepared by the double-decked paving method.” is very wide and cannot be positive from such experimental results, please review and seriously improve.
Round 2
Reviewer 2 Report
Not all the comments have been adequately addressed.
In particular:
- the double-decked paving method should be explain with more details
- it is not possible that the lower layer is at the same temperature of the upper layer during the double-decked compaction. Since some time passes between the laying of the lower layer and the laying of the upper layer, the lower layer will have a lower temperature when the upper layer is compacted. Please elaborate
- specific attention should be devoted to volumetrics (air voids of both layers, compaction profile, etc.) while no information is provided in this paper
- more details should be provided about the construction of the trial section (see comment in round 1)
- the use of English could be improved
Reviewer 3 Report
The use of English could be improved;
DOI numbers are missing;
